# Novel Tetrahydroisoquinoline-Based Heterocyclic Compounds Efficiently Inhibit SARS-CoV-2 Infection *In Vitro*

**DOI:** 10.3390/v15020502

**Published:** 2023-02-11

**Authors:** Xi Wang, Nikola T. Burdzhiev, Hengrui Hu, Yufeng Li, Jiang Li, Vesela V. Lozanova, Meglena I. Kandinska, Manli Wang

**Affiliations:** 1Center for Biosafety Mega-Science, Wuhan Institute of Virology, Chinese Academy of Sciences, Wuhan 430071, China; 2Faculty of Chemistry and Pharmacy, Sofia University “St. Kliment Ohridski”, 1, James Bourchier Avenue, 1164 Sofia, Bulgaria; 3Department of Medical Chemistry and Biochemistry, Medical Faculty, Medical University-Sofia, 2 Zdrave Str., 1431 Sofia, Bulgaria; 4Hubei Jiangxia Laboratory, Wuhan 430071, China

**Keywords:** SARS-CoV-2, antivirals, 1,2,3,4-tetrahydroisoquinoline, indole, chloroquine

## Abstract

The ongoing COVID-19 pandemic has caused over six million deaths and huge economic burdens worldwide. Antivirals against its causative agent, SARS-CoV-2, are in urgent demand. Previously, we reported that heterocylic compounds, i.e., chloroquine (CQ) and hydroxychloroquine (HCQ), are potent in inhibiting SARS-CoV-2 replication *in vitro*. In this study, we discussed the syntheses of two novel heterocylic compounds: *tert*-butyl *rel*-4-(((3*R*,4*S*)-3-(1*H*-indol-3-yl)-1-oxo-2-propyl-1,2,3,4-tetrahydroisoquinolin-4-yl)methyl)piperazine-1-carboxylate (*trans*-**1**) and *rel*-(3*R*,4*S*)-3-(1*H*-indol-3-yl)-4-(piperazin-1-ylmethyl)-2-propyl-3,4-dihydroisoquinolin-1(2*H*)-one (*trans*-**2**), which effectively suppressed authentic SARS-CoV-2 replication in Vero E6 cells. Compound *trans*-**1** showed higher anti-SARS-CoV-2 activity than *trans*-**2**, with a half maximal effective concentration (EC_50_) of 3.15 μM and a selective index (SI) exceeding 63.49, which demonstrated comparable potency to CQ or HCQ. Additional anti-SARS-CoV-2 tests on Calu-3 human lung cells showed that *trans*-**1** efficiently inhibited viral replication (EC_50_ = 2.78 μM; SI: > 71.94) and performed better than CQ (EC_50_ = 44.90 μM; SI = 2.94). The time of an addition assay showed that the action mechanism of *trans*-**1** differed from that of CQ, as it mainly inhibited the post-entry viral replication in both Vero E6 and Calu-3 cells. In addition, the differences between the antiviral mechanisms of these novel compounds and CQ were discussed.

## 1. Introduction

The COVID-19 pandemic caused by SARS-CoV-2 has resulted in more than 650 million human cases with an approximately 1% mortality (WHO data; 5 January 2023) [1]. Moreover, virus spillover to wild animals from people and transmission from animals such as pets to humans have been observed [2,3]; therefore, SARS-CoV-2 is expected to persist for a long period. As such, effective antivirals against SARS-CoV-2 are essential to human health care. Although there are a few approved small-molecule drugs for the treatment of COVID-19, such as Remdesivir [4], Paxlovid [5], and Molnupiravir [6], additional alternatives against different antiviral targets are needed in the wake of continuously emerging variants of SARS-CoV-2, especially for those with resistance to current drugs.

Heterocyclic compounds are a class of organic cyclic compounds with at least one hetero atom; most heteroatoms are nitrogen, sulfur, and oxygen [7]. These ring structures serve as the framework of many biological molecules such as DNA, RNA, hormones, and vitamins, making them indispensable motifs for drug discovery. Numerous FDA-approved drugs contain these heterocyclic structures, many of which are potent for the treatment of viral disease. For example, berberine inhibits the viral replication of herpes and Chikungunya viruses [8,9], as well as the entry of hepatitis C virus [10]. Moreover, it is active toward influenza virus *in vitro* and *in vivo* [11]. Isoquinoline alkaloids tetrandrine, fangchinoline, and cepharanthine hinder the expression of spike and nucleocapsid proteins in coronavirus OC43 in human lung cells [12]; palmatine inhibits the replication of West Nile, Zika, and Dengue viruses [13,14,15]; sanguinarine has antiviral effects against HIV protease and herpes simplex virus [16].

We previously reported that the anti-malarial drugs, i.e., chloroquine (CQ) and hydroxychloroquine (HCQ), are potent to inhibit SARS-CoV-2 replication *in vitro* [17,18]. One possible hypothesis for the inhibitory effect of CQ and HCQ is that the basicity of heterocycle in these compounds influences the acidity of intracellular organelles and hence membrane fusion [19] and the release of viral genetic materials into the cells. Heterocyclic compounds can potentially be modified to yield more optimized or varying bio-activities. For example, quinoline ring structure-based 1-oxo-2,3-disubstituted tetrahydroisoquinoline-4-carboxamides exhibit antiparasitic properties, particularly against *Plasmodium falciparum*, which is resistant to chloroquine treatment [20].

In this study, we synthesized two novel heterocyclic compounds based on the 1,2,3,4-tetrahydroisoquinoline structure and found that these compounds, i.e., *tert*-butyl *rel*-4-(((3*R*,4*S*)-3-(1*H*-indol-3-yl)-1-oxo-2-propyl-1,2,3,4-tetrahydroisoquinolin-4-yl)methyl)piperazine-1-carboxylate (*trans*-**1**) and *rel*-(3*R*,4*S*)-3-(1*H*-indol-3-yl)-4-(piperazin-1-ylmethyl)-2-propyl-3,4-dihydroisoquinolin-1(2*H*)-one (*trans*-**2**), effectively suppressed authentic SARS-CoV-2 infection in Vero E6 cell lines. Compound *trans*-**1** containing the Boc-protective group exhibited more optimized anti-SARS-CoV-2 activity. In addition, *trans*-**1** showed more pronounced antiviral activity than CQ in human lung Calu-3 cell lines. The mechanism of action of *trans*-**1** differs from that of CQ, which inhibits viral entry, as *trans*-**1** mainly functions at the post-entry stage. Our preliminary data on *trans*-**1** synthesized in this study suggest that it may be a candidate for fighting against COVID-19; however, additional data are needed to elucidate its clinical significance.

## 2. Materials and Methods

### 2.1. Cells and Viruses

African green monkey kidney Vero E6 epithelial cells were obtained from American Type Culture Collection (ATCC, NO. 1586) and maintained in an EMEM culture medium with 10% FBS. Human lung epithelial cells—Calu-3—were obtained from ATCC (HTB-55) and maintained in DMEM supplemented with 10% FBS. A clinically isolated SARS-CoV-2 strain WIV-04 [21] and the delta variant strain B.1.617.2 (IVCAS-6.7585) were obtained from the National Virus Resource Center (Wuhan, China) and were used for antiviral assessment following the approved standard operation procedures of Biosafety Level 3 (BSL-3) laboratory at Wuhan Institute of Virology, Chinese Academy of Sciences.

### 2.2. Chemical Synthesis of Novel Heterocyclic Compounds

All solvents used in the present study were of HPLC grade and are commercially available. We synthesized the starting tosylate in previous studies [22]; 1-Boc-piperazine and trifluoroacetic acid are commercially available and were used as supplied. The melting points (m.p.) of the compounds were determined on Boetius PHMK 0.5 apparatus and were uncorrected. NMR spectra were obtained using a Bruker Avance III 500 HD NMR spectrometer operating at 500.13 MHz for ^1^H and 125.76 MHz for ^13^C NMR. The chemical shifts are presented in ppm (δ) using tetramethylsilane (TMS) as an internal standard. Liquid chromatography mass spectrometry analysis (LC-MS) was carried out on a Q Ex-active^®^ hybrid quadrupole-Orbitrap^®^ mass spectrometer (ThermoScientific Co, Waltham, MA, USA) equipped with a HESI^®^ (heated electrospray ionization) module, a Tur-boFlow^®^ Ultra High Performance Liquid Chromatography (UHPLC) system (Thermo-Scientific Co, Waltham, MA, USA), and an HTC PAL^®^ autosampler (CTC Analytics, Zwingen, Switzerland). The synthetic procedures for *trans-***1** and *trans-***2** were as follows.

#### 2.2.1. Synthesis of *tert*-butyl *rel*-4-(((3*R*,4*S*)-3-(1*H*-indol-3-yl)-1-oxo-2-propyl-1,2,3,4-tetrahydroisoquinolin-4-yl)methyl)piperazine-1-carboxylate (*trans*-**1**)

A mixture of tosylate (0.489 g, 1 mmol) and 1-Boc-piperazine (0.559 g, 3 mmol) in dry toluene (5 mL) was refluxed, until the starting tosylate reacted completely (TLC). The reaction mixture was cooled down to 25 °C, and crystals were formed, filtered and discarded. The filtrate was concentrated, and another portion of crystals was collected. The solvent from the filtrate containing the desired product was evaporated under reduced pressure, and the residual oil was crystallized in ethyl acetate (2 mL) and light petroleum (initially 6 mL). An additional 3 mL of light petroleum was added upon crystal formation. Crystals were collected by filtration and dried, thus yielding 0.371 g (73.8%) of white crystals.

m.p. 102–104 °C; ^1^H NMR (CDCl_3_, 500.13 MHz) δ 0.96 (t, 3H, CH_2_C*H*_3_, *J* = 7.4 Hz), 1.50 (s, 9H, C(C*H*_3_)_3_), 1.71 (sext, 2H, C*H*_2_CH_3_, *J* = 7.4 Hz), 2.30–2.50 (m, 3H, 1xCHC*H*_2_, 2xC*H*_2_-pip), 2.60–2.90 (m, 4H, 1xNC*H*_2_, 1xCHC*H*_2_, 2xC*H*_2_-pip), 3.29 (dd, 1H, H-4, *J* = 3.6, 11.7 Hz), 3.55–3.70 (m, 4H, C*H*_2_-pip), 4.19 (ddd, 1H, NC*H*_2_, *J* = 8.1, 8.1, 13.7 Hz), 5.59 (br.s, 1H, H-3), 6.62 (d, 1H, C*H*N-Ind, *J* = 1.5 Hz), 6.91–6.95 (m, 1H, H-5), 7.16 (dd, 1H, C*H*-Ind, J = 7.4, 7.7 Hz), 7.20 (dd, 1H, C*H*-Ind, *J* = 7.4, 7.4 Hz), 7.28–7.36 (m, 3H, C*H*-Ind, H-6, H-7), 7.54 (d, 1H, C*H*-Ind, *J* = 7.7 Hz), 8.12–8.17 (m, 1H, H-8), 8.22 (br. s, 1H, NH); ^13^C NMR (CDCl_3_, 125.76 MHz) δ 11.67 (1C, CH_2_*C*H_3_), 21.48 (1C, *C*H_2_CH_3_), 28.46 (3C, C(*C*H_3_)_3_), 42.31 (1C, C-4), 43.50 (1C, *C*H_2_-pip), 44.23 (1C, *C*H_2_-pip), 47.63 (1C, *C*H_2_N), 53.21 (1C, C-3), 53.43 (2C, *C*H_2_-pip), 62.19 (1C, CH*C*H_2_), 79.85 (1C, *C*(CH_3_)_3_), 111.66 (1C, CH-Ind), 115.64 (1C, C-Ind), 117.57 (1C, CH-Ind), 119.73 (1C, CH-Ind), 121.96 (1C, CHN-Ind), 122.28 (1C, CH-Ind), 125.85 (1C, C-Ind); 127.31 (1C, C-6), 128.00 (1C, C-8), 128.35 (1C, C-5), 128.79 (1C, C-8a), 131.79 (1C, C-7), 136.37 (1C, C-Ind), 138.36 (1C, C-4a), 154.91 (1C, NCOO), 163.80 (1C, C-1). ESI-HRMS (m/z) calculated for [M + H]^+^ ion species C_30_H_39_N_4_O_3_: 503.3022; found: 503.3075.

#### 2.2.2. Synthesis of rel-(3R,4S)-3-(1H-indol-3-yl)-4-(piperazin-1-ylmethyl)-2-propyl-3,4-dihydroisoquinolin-1(2H)-one (trans-2)

Compound *trans*-**1** (0.402 g, 8 mmol) was dissolved in trifluoroacetic acid (2 mL) and sonicated for 15 min. The reaction mixture was concentrated under reduced pressure, and the oily residue was triturated with 10% Na_2_CO_3_ solution (6 mL) upon which crystals were formed. The solid was collected by filtration to yield 0.123 g of white crystals. The filtrate was extracted with ethyl acetate (3 × 10 mL), dried with Na_2_SO_4_ and evaporated to obtain an additional 0.068 g of white crystals for a combined yield of 0.191 g (60%).

m.p. 123–125 °C; ^1^H NMR (CDCl_3_, 500.13 MHz) δ 0.96 (t, 3H, CH_2_C*H*_3_, *J* = 7.4 Hz), 1.69 (sext, 2H, C*H*_2_CH_3_, *J* = 7.4 Hz), 2.45 (dd, 1H, CHC*H*_2_, *J* = 4.3, 12.9 Hz), 2.59–2.70 (m, 2H, C*H*_2_-pip), 2.75–2.88 (m, 2H, 1xNC*H*_2_, 1xCHC*H*_2_), 2.88–3.00 (m, 2H, C*H*_2_-pip), 3.22–3.31 (m, 5H, H-4, C*H*_2_-pip), 4.11–4.21 (m, 1H, NC*H*_2_),4.46 (br.s, 1H, NH), 5.48 (s, 1H, H-3), 6.62 (br.s, 1H, C*H*N-Ind), 6.90–6.95 (m, 1H, H-5), 7.16 (dd, 1H, C*H*-Ind, J = 7.4, 7.7 Hz), 7.21 (dd, 1H, C*H*-Ind, *J* = 7.4, 7.4 Hz), 7.30–7.37 (m, 3H, C*H*-Ind, H-6, H-7), 7.48 (d, 1H, C*H*-Ind, *J* = 7.7 Hz), 8.12–8.18 (m, 2H, H-8, NH-Ind); ^13^C NMR (CDCl_3_, 125.76 MHz) δ 11.66 (1C, CH_2_*C*H_3_), 21.54 (1C, *C*H_2_CH_3_), 42.18 (1C, C-4), 44.62 (2C, *C*H_2_-pip), 47.70 (1C, *C*H_2_N), 51.87 (2C, *C*H_2_-pip), 53.38 (1C, C-3), 62.24 (1C, CH*C*H_2_), 111.73 (1C, CH-Ind), 115.44 (1C, C-Ind), 117.37 (1C, CH-Ind), 119.84 (1C, CH-Ind), 122.02 (1C, CHN-Ind), 122.39 (1C, CH-Ind), 125.75 (1C, C-Ind); 127.49 (1C, C-6), 128.08 (1C, C-8), 128.37 (1C, C-5), 128.76 (1C, C-8a), 131.89 (1C, C-7), 136.34 (1C, C-Ind), 137.99 (1C, C-4a), 163.73 (1C, C-1). ESI-HRMS (m/z) calculated for [M + H]^+^ ion species C_25_H_31_N_4_O: 403.2498; found: 403.2622.

### 2.3. Antiviral Assessment

Compounds were dissolved in dimethyl sulfoxide (DMSO) and diluted to a working concentration with a cell culture medium. Cells were incubated with compounds following infection with SARS-CoV-2 (MOI = 0.05) for 24 h or 48 h for Vero E6 and Calu-3 cells, respectively. The antiviral effect was measured using two previously reported approaches [17]: (1) quantification of the cell supernatant to assess progeny virus yield; and (2) immunofluorescence staining of the viral nucleocapsid protein (NP) to illustrate the infected cells. For quantitative real-time RT-PCR (qRT-PCR), the viral RNA was extracted by a commercial kit (Takara Bio, Beijing, China, #9766) and reverse-transcribed with a PrimeScript RT Reagent Kit (Takara Bio, Beijing, China, #RR047B). qRT-PCR was performed on StepOne Plus Real-time PCR system (Applied Biosystem) with TB Green Premix Ex Taq II (Takara Bio, Beijing, China, #RR820A). The primers used for qRT-PCR were RBD-qF: 5′-CAATGGTTTAACAGGCACAGG-3′ and RBD-qR:5′-CTCAAGTGTCTGTGGATCACG-3′.

### 2.4. Cytotoxicity of Tested Compounds

To evaluate the cytotoxicity of compounds, a series of diluted concentrations of compounds were incubated with Vero E6 or Calu-3 cells in a 96-well plate (1 × 10^4^ cells/well) for 24 h, following cell viability assessment with a cell count kit-8 (Beyotime, Shanghai, China, #C0039) according to the manufacturer’s instructions. The OD_450_ values of the compound treated cells (OD_compound_), DMSO-treated cells (OD_DMSO_), and a medium without cells (OD_blank_) were obtained for calculating the normalized cytotoxicity under each concentration, which was expressed as: % cytotoxicity = 100% − (OD_compound_ − OD_blank_) / (OD_DMSO_ − OD_blank_) × 100%. The CC_50_ for each compound on each cell type was calculated using Graphpad Prism 8.0 software.

### 2.5. Time of the Drug Addition Assay

To determine the point where *trans*-**1** inhibited viral replication, a time of an additional assay was performed using three treatments: “Full-time”, “Entry”, and “Post-entry”. For the “Entry” treatment, cells were pretreated with *trans*-**1** (20 μM) for 1 h and then infected with SARS-CoV-2 (MOI = 0.05). After incubation for 1 h, compound *trans*-**1** containing a medium was removed and replaced in a fresh medium after washing once with PBS, and cells were cultured for an additional 24 or 48 h for Vero E6 and Calu-3 cells, respectively. For the “Post-entry” treatment, the cells were incubated with viruses for 1 h, after which the cells were washed with PBS and *trans*-**1** (20 μM) containing a medium was added. For the “Full-time” treatment, cells were treated with *trans*-**1** before, during, and after virus incubation. Samples were collected in three ways: 1) the cell supernatant was collected for virus production detection; 2) infected cells were fixed with 4% *w*/*v* Paraformaldehyde (PFA) for immunofluorescence assay; and 3) infected cells were lysed with 1 × SDS-PAGE loading buffer (50 mM Tris-HCl, 2% *w*/*v* SDS, 0.1% *w*/*v* bromophenol blue, 10% *v*/*v* glycerol, and 1% *v*/*v* β-mercaptoethanol) for Western blot analysis.

### 2.6. Immunofluorescence Assay

Cells were fixed in 4% PFA for 24 h to completely inactivate residual viruses, then further permeabilized with 0.2% *v*/*v* Triton and blocked with 5% *w*/*v* BSA. A mouse monoclonal antibody against the NP (Wuhan Keyuan Ambo Biotechnology Co. LTD., Wuhan, China, #AB0001, 1 μg/mL) was incubated with cells for 1 h, and after extensive washing with PBS, an Alexa-488-labeled goat antimouse antibody (Abcam, Cambridge, UK, #ab150113, 1:500) was incubated for 1 h. Nuclei were stained with Hoechst 33258 (Beyotime, Shanghai, China). Fluorescent signals were observed using fluorescence microscopy.

### 2.7. Western Blot Analysis

Protein samples were subjected to 12% SDS-PAGE and then transblotted onto polyvinylidene difluoride (PVDF) membranes (Millipore). After blocking with 5% *m*/*v* milk in TBS buffer, the membrane was probed with the mouse mAb anti-NP (Wuhan Keyuan Ambo Biotechnology Co. LTD., Wuhan, China, #AB0001, 2 μg/mL) and the horseradish peroxidase (HRP)-conjugated Goat Anti-Mouse IgG (Proteintech, Chicago, IL, USA, #SA00001-1, 1:5000) as the primary and the secondary antibody, respectively. Protein bands were detected using a SuperSignal West Pico Chemiluminescent substrate (Pierce).

## 3. Results

### 3.1. Synthesis of Novel Tetrahydroisoquinoline-Based Heterocyclic Compounds

Based on the results of the biological activity of the previously synthesized 1,2,3,4-tetrahydroisoquinoline (THIQ) derivatives and piperidinones, additional heterocyclic moieties to the THIQ core structure were selected. The synthesis of *trans*-**1** was based on the known in the literature reaction scheme (Figure 1A) [22]. Compound *trans*-**2** was obtained from isoquinoline *trans*-**1** after the removal of the Boc- protective group in trifluoroacetic acid (Figure 1B).

### 3.2. Anti-SARS-CoV-2 Activity of Novel Heterocyclic Compounds

The anti-SARS-CoV-2 potentials of the two THIQ derivatives *trans*-**1** and *trans*-**2** and the seven additional heterocyclic compounds previously synthesized by us [23] were tested on Vero E6 cells. Compounds *trans*-**3**–**9** shared the same main heterocyclic moieties as *trans*-**1** and *trans*-**2**, and their general structures are displayed in Appendix A. Two dosages of each compound (20 μM and 5 μM) were subjected to Vero E6 cell culture, following viral infection with an MOI of 0.05. Chloroquine (CQ) (10 μM) was used as an inhibitory positive control. The supernatant of cell culture was harvested at 24 h p.i. for measuring progeny virus production by qRT-PCR. As shown in Figure 2, *trans*-**1** (both 20 μM and 5 μM) and *trans*-**2** (20 μM) with the indicated concentration significantly decreased the production of viral progeny more than 10-fold (Figure 2A). The immunofluorescence staining of the viral NP also suggested that *trans*-**1** and *trans*-**2** with the indicated concentration inhibited the replication of SARS-CoV-2 (Figure 2B).

Subsequently, we determined the EC_50_ and CC_50_ values of the two compounds on Vero E6 cells (Figure 3) by measuring the viral progeny load in a culture supernatant. Compound *trans*-**1** showed limited cytotoxicity as the CC_50_ value exceeded the highest tested concentration (200 μM) (Figure 3A), while the CC_50_ value of *trans*-**2** was 67.78 μM (Figure 3C). The EC_50_ values of *trans*-**1** and *trans*-**2** were 3.15 μM and 12.02 μM, respectively, and their SI values were >63.49 and 5.64, respectively (Figure 3A,C). The infected cells indicated by the viral NP staining showed the dose-dependent antiviral effect of each compound (Figure 3B,D). Based on these results, two kinds of the novel heterocyclic compounds, *trans*-**1** and *trans*-**2**, showed anti-SARS-CoV-2 potency on Vero E6 cells, and *trans*-**1** was superior to *trans*-**2**.

### 3.3. Compound Trans-**1** Efficiently Inhibits SARS-CoV-2 Replication in Human Lung Cells

CQ was found to have limited potency in inhibiting viral infection of human lung cells that express co-receptor TMPRSS2, such as Calu-3, which facilitates S protein priming and mediates viral entry via plasma membrane fusion instead of endocytosis [24]. We wondered whether the heterocyclic compound developed in this study, particularly the most promising candidate *trans*-**1**, could inhibit SARS-CoV-2 infection in human lung cells. As shown in Figure 4A,B, surprisingly, *trans*-**1** (10 μM) inhibited SARS-CoV-2 Delta variant (B.1.617.2) infection of Calu-3, which was not the case for CQ (Figure 4A,B). The EC_50_ values for *trans*-**1** and CQ in Calu-3 were 2.78 μM and 44.9 μM, respectively (Figure 4C). Therefore, *trans*-**1**, a novel heterocyclic compound, may have a unique antiviral mechanism that effectively blocks SARS-CoV-2 infection in human lung cells.

### 3.4. Mode of Action of Trans-**1**

As CQ inhibits virus endocytosis and, as mentioned above, *trans*-**1** probably has a different antiviral mechanism, we tested whether *trans*-**1** also inhibits viral entry. A time of addition assay was performed. As shown in Figure 5A, *trans*-**1** inhibited the production of viral progeny by almost 100% in both the “Full-time” and “Post-entry” treatment groups, and it only inhibited the production of viral progeny by ~20% and ~10% in the “Entry” treatment on Vero E6 and Calu-3 cells, respectively. The NP expression detected by Western blot (Figure 5B) and immunofluorescence (Figure 5C) also confirmed the inhibitory effect in the “Full-time” and “Post-entry” stages. Therefore, *trans*-**1** inhibits SARS-CoV-2 replication mainly at the post-entry stage, which differs from CQ.

## 4. Discussion

Piperazine, alkylpiperazine, and phenylpiperazine are important heterocyclic building blocks extensively evaluated in the development of various biologically active agents including anticonvulsants, antidepressants, antimalarials, anti-HIV compounds [25], and carbonic anhydrase inhibitors [26]. We previously reported that the heterocyclic drug CQ [17], HCQ [18], and other anti-malaria drugs such as arteannuin B [27] exhibit potency in the inhibition of SARS-CoV-2 infection *in vitro*. In this study, we developed two novel tetrahydroisoquinoline-based heterocyclic compounds (Figure 1 and Supporting Information) and tested their anti-SARS-CoV-2 activities (Figure 2). Both *trans*-**1** and *trans*-**2** effectively inhibited viral infection, with EC_50_ values of 3.15 μM and 12.02 μM, respectively (Figure 3A,C). Compounds *trans*-**1** and *trans*-**2** had similar foundations, and the removal of the Boc-protection of amino group in *trans*-**2** was the only difference. Therefore, considering that *trans*-**2** was less biologically active against SARS-CoV-2, the *N*-Boc-piperazine group in the 4th place appeared to be crucial for the anti-viral activity. This protection group also seemed to reduce compound cytotoxicity to cells as the CC_50_ of *trans*-**2** was 67.78 μM (Figure 3C), whereas low cytotoxicity was observed for *trans*-**1** even at 200 μM (Figure 3A and Figure 4C).

Alkaloids, CQ, and HCQ could elevate the pH of endosome, thus inhibiting viral entry which demands low endosomal pH [28]. The cleavage of S1/S2 of spike protein is essential for SARS-CoV-2 entry and could be mediated by endosomal cathepsin B/L or TMPRSS2 located on the cell surface [29]. Because the activity of endosomal cathepsin relies on acid pH, CQ effectively suppresses the endosomal entry of SARS-CoV-2 in TMPRSS2 negative cells [18]. However, for TMPRSS2-expressing cells such as respiratory epithelial cells and lung cells (such as Calu-3), CQ showed limited antiviral potency [24]. This may explain the limited clinical anti-SARS-CoV-2 efficacy of CQ and HCQ [30]. Because *trans*-**1** is the most promising compound we have obtained so far and our starting strategy to synthesize novel heterocyclic compounds was based on CQ’s antiviral activity, we attempted to dissect the working mechanism of *trans*-**1**. First, we tested the anti-SARS-CoV-2 activities of *trans*-**1**, *trans*-**2**, and CQ on Calu-3 cells. As shown in Figure 4, only *trans*-**1** showed potency in the inhibition of viral infection under 10 μM, and the EC_50_ value of *trans*-**1** was considerably higher than that of CQ (2.78 μM versus 44.9 μM). This result was unexpected and suggested that *trans*-**1** might possess a unique mechanism for the inhibition of viral replication. Therefore, a time of addition assay was performed to reveal whether *trans*-**1** functions at the viral entry stage in the same way CQ does. The results showed that *trans*-**1** inhibited SARS-CoV-2 replication mainly at the post-entry stage in both Vero E6 and Calu-3 cells (Figure 5). Compound *trans*-**1** probably targets the host cell for antiviral activity, but this target might not be associated with pH increase in endosome as CQ does. Of course, we do not exclude the possibility that *trans*-**1** may directly target viral machines to inhibit virus replication in host cells. Taken together, due to different working mechanisms, *trans*-**1** is promising in that it potently inhibits SARS-CoV-2 infection in human lung cells, while CQ and HCQ fail to.

In this study, *trans*-**1** was shown with potency to inhibit WIV-04 and Delta SARS-CoV-2 strains. If *trans*-**1** targets the host cell biological events, it would be potent in the inhibition of most variants of concern, including ongoing endemic Omicron strains. Although the *in vitro* anti-SARS-CoV-2 activity of *trans*-**1** is not as good as current clinically used therapeutics such as Molnupiravir and Paxlovid, the *in vivo* antiviral efficacy is a more important parameter and it remains to be determined. Thus, additional preclinical data are needed to evaluate whether *trans*-**1** is a promising candidate for the treatment of COVID-19 and the consequent mitigation of the pandemic, including the detailed anti-viral mechanism of *trans*-**1** and the *in vivo* potency in the inhibition of SARS-CoV-2 infection of proper animal models. In addition, based on the structural formulations of *trans*-**1** and *trans*-**2**, more analogs have been synthesized, and their anti-coronavirus activity as well as cytotoxicity are being tested. Therefore, a comprehensive view of the relationship between the structure and the bio-activity of these THIQ analogs would offer insights into a more optimized design of heterocyclic compounds with a broad spectrum of anti-coronavirus activity in the future.

## Figures and Tables

**Figure 1 viruses-15-00502-f001:**
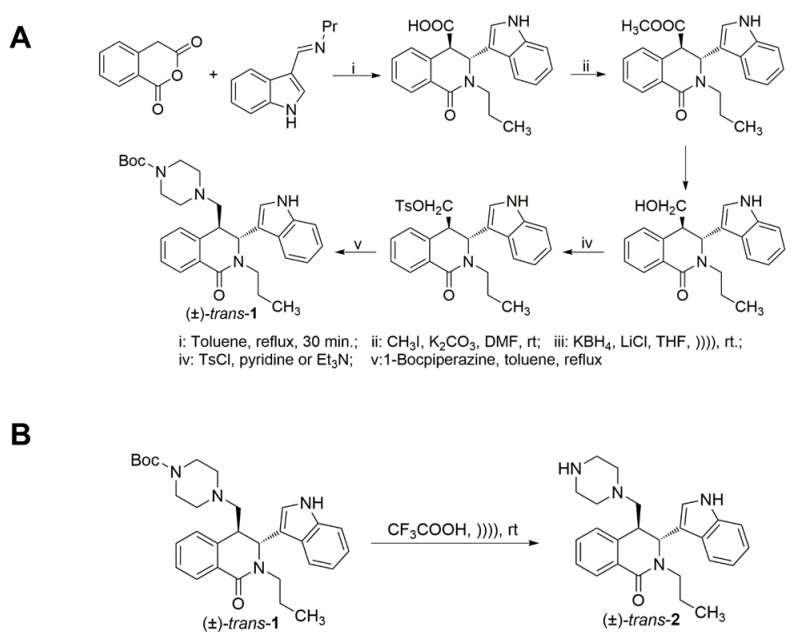
Representative chemical synthesis routine of active compounds *trans*-**1** and *trans*-**2**. (**A**) Synthetic strategy of *trans*-**1**. The reagents and conditions for each step numbered i to v are presented below the scheme. (**B**) The BOC−piperazine group of *trans*-**1** was removed to obtain *trans*-**2**.

**Figure 2 viruses-15-00502-f002:**
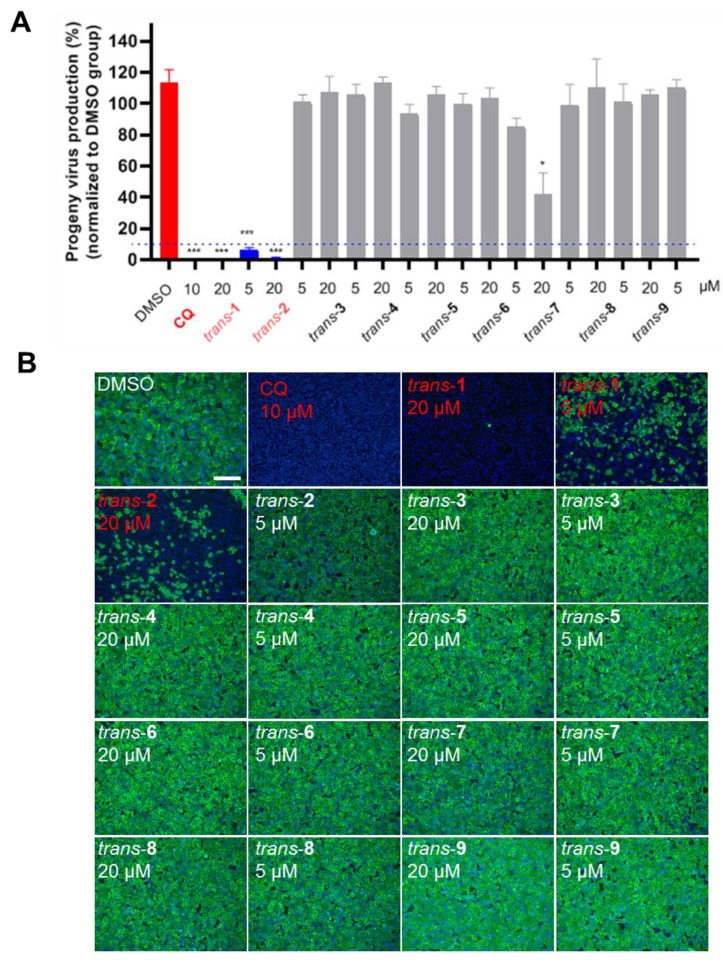
Screening for compounds with anti-SARS-CoV-2 activity. Vero E6 cells were treated with 20 μM or 5 μM of each compound and infected with SARS-CoV-2 (MOI = 0.05). (**A**) The supernatant was collected 24 h post-infection (hpi), and virus production was quantified using qRT-PCR. *trans*-**1** and *trans*-**2** were labeled in a red font, as at least one of the working concentrations decreased the viral load by ≥90% (blue dashed line). Vehicle (DMSO) and chloroquine (10 μM) treatments were used as a negative and positive control, respectively. Each concentration of all compounds was performed in three replicates. A *t* test was performed to show the differences between each group and DMSO control. * indicates *p* < 0.05; *** indicates *p* < 0.001. (**B**) IFA results of screening for compounds with anti-SARS-CoV-2 activity. The infected cells were fixed with 4% PFA, and an immune staining of a viral NP was used to show the viral positive cells. Four treatments, i.e., 10 μM CQ, 20 μM *trans*-**1**, 5 μM *trans*-**1**, and 20 μM *trans*-**2**, inhibited viral infection obviously, and they are labeled in a red font. The scale bar is 400 μm.

**Figure 3 viruses-15-00502-f003:**
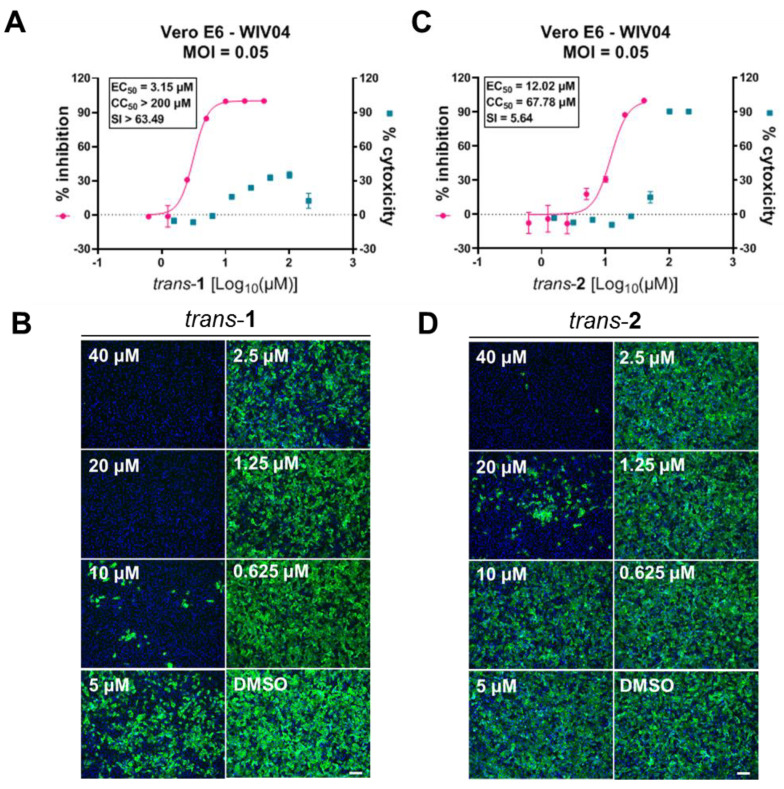
EC_50_ values of *trans*-**1** and *trans*-**2** on Vero E6 cells. Vero E6 cells were treated with the indicated concentrations of *trans*-**1** (**A**,**B**) or *trans*-**2** (**C**,**D**) and were infected with SARS-CoV-2 at an MOI of 0.05. At 24 h p.i., the supernatant was collected to determine the production of viral progeny by qRT-PCR, and the inhibition rate of viral production was calculated. The cytotoxicity values of each compound under the indicated concentrations were measured using a CCK-8 assay. The dose−response curves were plotted using GraphPad Prism 8 (**A**,**C**). The infected cells were fixed with 4% PFA, and an immune staining of the viral NP was used to show the viral positive cells (**B**,**D**). The scale bar for each immunofluorescence image is 200 μm.

**Figure 4 viruses-15-00502-f004:**
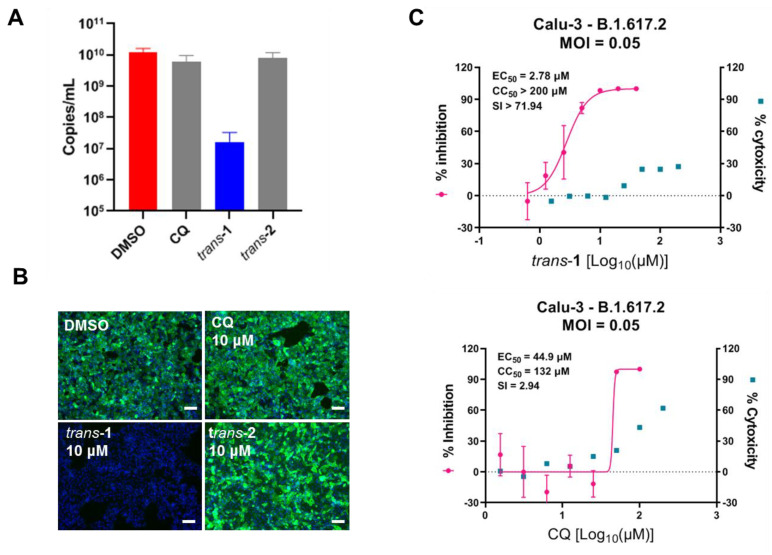
Compound *trans*-**1** exhibits potency in the inhibition of SARS-CoV-2 replication in Calu-3 cells. A single dose of 10 μM of CQ, *trans*-**1**, or *trans*-**2**, was used to treat Calu-3 cells, followed by SARS-CoV-2 (B.1.617.2) infection at an MOI of 0.05 for 48 h. The supernatant and the infected cells were harvested to measure the production of viral progeny (**A**) and the immune staining of the NP (**B**), respectively. The scale bar of immunofluorescence image in (**B**) is 200 μm. (**C**) Dose-dependent effects of *trans*-**1** and CQ on SARS-CoV-2 (B.1.617.2) on Calu-3 cells. Calu-3 cells were treated with the indicated concentration of *trans*-**1** or CQ and were infected with SARS-CoV-2 (B.1.167.2) at an MOI of 0.05 for 48 h. The supernatant was quantified for the production of viral progeny, and the inhibition rate was normalized to that of the DMSO vehicle control group. Combined with the cytotoxicity data of each compound, the dose−response curves of the inhibition rate and cytotoxicity were generated using GraphPad Prism 8 software.

**Figure 5 viruses-15-00502-f005:**
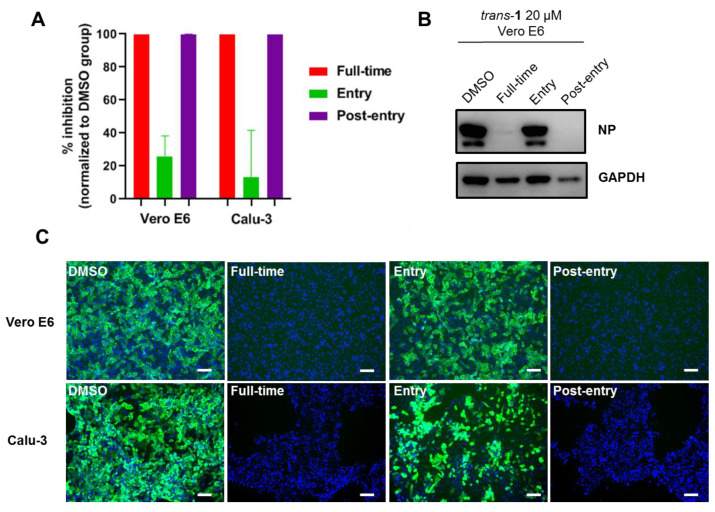
Compound *trans*-**1** inhibits SARS-CoV-2 replication at the post-entry stage. A time of addition assay was performed to determine the point where *trans*-**1** inhibits viral infection. For the “Entry” treatment, the Vero E6 or Calu-3 cells were pre-incubated with *trans*-**1** (20 μM) containing a medium for 1 h and infected with SARS-CoV-2 (B.1.617.2) at an MOI of 0.05 for 1 h. Afterwards, the supernatant was removed, and the cells were washed with PBS and incubated with *trans*-**1**-free medium. For the “post-entry” treatment, a *trans*-**1** containing medium was only added after viral infection for 1 h. For the “full-time” treatment, *trans*-**1** was constantly present. Three kinds of samples were collected at the end of the experiment (24 h p.i. for Vero E6; 48 h p.i. for Calu-3): the supernatant was harvested for the quantification of the virus yield and the calculation of the inhibition rate (**A**); infected cells were lysed with 1 × SDS-PAGE loading buffer and analyzed for the NP expression via Western blotting (**B**); infected cells were fixed with 4% PFA, and immunofluorescence assay was performed to show the viral NP positive cells (**C**). The scale bar is 100 μm.

## Data Availability

The data are available within the text.

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
