# Peer review of "Novel Tetrahydroisoquinoline-Based Heterocyclic Compounds Efficiently Inhibit SARS-CoV-2 Infection In Vitro"

_viruses, 2023, doi:10.3390/v15020502_

Round 1

Reviewer 1 Report

The paper described the development of two novel tetrahydroisoquinoline based heterocyclic compounds (trans-1 and trans-2) and tested their in-vitro anti-SARS-CoV-2 activity. The findings are significant in that trans-1 has a good SI (>71.9) and has the potency to inhibit most variants of concern, probably by targeting host cell biological events. Overall, the study is well designed and performed, the conclusion has clinical potential, and the manuscript is well written and organized. Several minor revisions are required to improve the study:

1) line 73, I suggest to use another phase to replace“make a credible conclusion”, such as “elucidate its clinical significance”.

2) line 109 and line 132, the word“m.p.”is confusing for most readers, please add several words to explain it and its following parameters.

3) line 174, catalog number of the mAb against NP is needed.

4) line 213, I suggest add“at least”before“one of the working concentration”, as trans-1 can decrease viral load >90% at both 20uM and 5uM.

5) line 265-267, “As”,“so”and ”Therefore”seem to be redundant.

6) line 284, is it“1XSDS-PAGE loading buffer”or“lysis buffer”? 

7) line 286, scale bar of panel C is missing.

Author Response

The paper described the development of two novel tetrahydroisoquinoline based heterocyclic compounds (trans-1 and trans-2) and tested their in-vitro anti-SARS-CoV-2 activity. The findings are significant in that trans-1 has a good SI (>71.9) and has the potency to inhibit most variants of concern, probably by targeting host cell biological events. Overall, the study is well designed and performed, the conclusion has clinical potential, and the manuscript is well written and organized. Several minor revisions are required to improve the study:

Response 1: We thank the reviewer for the positive comments and valuable suggestions.

1) line 73, I suggest to use another phase to replace “make a credible conclusion”, such as “elucidate its clinical significance”.

 Response 2: It has been revised according to the reviewer’s suggestion (line 74).

2) line 109 and line 132, the word “m.p.”is confusing for most readers, please add several words to explain it and its following parameters.

 Response 3: The word “m.p.” is the abbreviation of “melting point”. The abbreviation has been added in “Materials and Methods” Section (line 89).

3) line 174, catalog number of the mAb against NP is needed.

  Response 4: The antibody information has been added (line 186).

4) line 213, I suggest add “at least” before “one of the working concentration”, as trans-1 can decrease viral load >90% at both 20uM and 5uM.

 Response 5: Corrected (line 228).

5) line 265-267, “As”,“so”and ”Therefore”seem to be redundant.

 Response 6: We deleted “therefore” in line 282.

6) line 284, is it“1XSDS-PAGE loading buffer”or“lysis buffer”? 

 Response 7: It’s 1×SDS-PAGE loading buffer, and its formula (50 mM Tris-HCl, 2% w/v SDS, 0.1% w/v bromophenol blue, 10% v/v glycerol, 1% v/v β-mercaptoethanol) has been explained in line 180-181.

7) line 286, scale bar of panel C is missing.

Response 8: The scale bar for images in Fig. 5C is 100 μm, and it has been added in line 301.

Reviewer 2 Report

The manuscript entitled: Novel tetrahydroisoquinoline-based heterocyclic compounds efficiently inhibit SARS-CoV-2 infection in vitro by Xi Wng, and the team reported the synthesis and mechanistic details of antiviral activity of two novel heterocyclic compounds,  which effectively suppressed SARS-CoV-2 replication in vitro. The study is interesting; however, there are a few major concerns. 

Although Chloroquine and Hydroxychloroquine showed promising results initially, they have been heavily criticized for their use in humans in recent times. Hence, the central question is, how the novel heterocyclic compounds will be different down the line? How is this study novel and addressing the drawback of CQ and HCQ? And, more importantly, how these molecules are superior to the current clinically used therapeutics. Authors need to significantly modify the discussion (and introductions) accordingly. 

Authors should consider including the data for purity of the synthesized derivatives in the main manuscript. 

Section 2.4: Briefly describe the method. Also, which kit was used? 

Section 2.6/2.7 should have all the vital details including antibody catalog number, dilution, and incubation for all the antibodies used in the study discussed in this manuscript.

Authors should include another control molecule that is being clinically used to get a better idea of these novel derivatives' antiviral potential compared to existing therapeutics.

It will be important to determine the antiviral efficacy of trans-1 and trans-2 at different MOI. 

How effective are these compounds on the other more virulent strains of SARS-CoV-2?

Fig.2 A: The methods section must have qRT-PCR details. What primers were used? 

Fig.5 B: How many times the western blot was performed? The internal control of GAPDH is significantly unequal. Hence, a better-quality image or a bar graph with densitometric analysis will be required.  

Author Response

The manuscript entitled: Novel tetrahydroisoquinoline-based heterocyclic compounds efficiently inhibit SARS-CoV-2 infection in vitro by Xi Wang, and the team reported the synthesis and mechanistic details of antiviral activity of two novel heterocyclic compounds, which effectively suppressed SARS-CoV-2 replication in vitro. The study is interesting; however, there are a few major concerns. 

Response 9: We thank the reviewer for the careful review and helpful comments

Although Chloroquine and Hydroxychloroquine showed promising results initially, they have been heavily criticized for their use in humans in recent times. Hence, the central question is, how the novel heterocyclic compounds will be different down the line? How is this study novel and addressing the drawback of CQ and HCQ? And, more importantly, how these molecules are superior to the current clinically used therapeutics. Authors need to significantly modify the discussion (and introductions) accordingly. 

Response 10: We see the concern of the reviewer. Although sharing with some structural similarity (isoquinoline ring structure), a promising feature of trans-1 is that it potently inhibits SARS-CoV-2 infection in human lung cells, while CQ and HCQ fail to. The mechanism of trans-1 (targets post entry events) seems to be also different from CQ/HCQ (inhibits virus entry). Therefore, trans-1 represents a novel type of anti-SARS-CoV-2 inhibitor. We have compared the in vitro anti-SARS-CoV-2 activity of trans-1 with EIDD-1931 (the active form of Monulpiravir) in parallel study, and the EC50 value of trans-1 is higher than that of EIDD-1931 (please see Response 14). However, in vivo antiviral efficacy is a more important factor and it remains to be determined for trans-1. In addition, the study provided useful references for further designing and optimization of novel isoquinoline-based heterocyclic compounds against SARS-CoV-2 and probably other viruses. We have added relevant discussion in lines 340-342, 345-348.

Authors should consider including the data for purity of the synthesized derivatives in the main manuscript. 

Response 11: The analytical purity of the compounds is confirmed by HRMS analysis data and NMR as well. HRMS and NMR spectra can be found in the Supplementary Figures 1-6.

Section 2.4: Briefly describe the method. Also, which kit was used? 

Response 12: The detailed method and the kit used for cytotoxicity measurement has been added (line 162-166).

Section 2.6/2.7 should have all the vital details including antibody catalog number, dilution, and incubation for all the antibodies used in the study discussed in this manuscript.

Response 13: The related information has been added (line 186-188 and 194-196).

Authors should include another control molecule that is being clinically used to get a better idea of these novel derivatives' antiviral potential compared to existing therapeutics.

Response 14: We see the reviewer’s point. We actually compared the antiviral efficiency of trans-1 and EIDD-1931 (the active form of Molnupiravir) in parallel experiments. The EC50 value of EIDD-1931 on Calu-3 cells was 0.23 μM, about ten times lower than that of trans-1. EIDD-1931 is a nucleoside analog and showed different mode of action from CQ/trans-1. In the prevention and treatment of emerging mutant strains, various types of antiviral drugs need to be developed. The purpose of this work is to optimize the research based on heterocyclic compound skeleton, so as to provide reference for the design of new inhibitors against SARS-CoV-2 and even other viruses. Of course, the in vivo effecacy of trans-1 remains to be tested in the future.

It will be important to determine the antiviral efficacy of trans-1 and trans-2 at different MOI. 

Response 15: Indeed, different EC50 values might be obtained even for the same compound when virus infections are performed at different MOIs, as we previously reported on CQ and HCQ (Liu et al. 2020. PMID 32194981). We found that MOI of 0.05 is a commonly used condition for testing compounds against SARS-CoV-2. To better compare with other drugs, we also used MOI of 0.05 in this study.

How effective are these compounds on the other more virulent strains of SARS-CoV-2?

Response 16: Due to the limitation of virus strain source and the availability of BSL-3 laboratory, we have not tested other SARS-CoV-2 variants except B.1.617.2 (Delta) strain. We found that trans-1 mainly functions at post-entry stage of virus infection instead of targeting virus entry (mediated by viral spike protein). Therefore, we expect trans-1 is likely to be also effective against other SARS-CoV-2 strains (mainly attributed to spike protein mutations), which can be partially supported by its activity against B.1.617.2 (Delta) strain. We also agree with the reviewer that it is meaningful to test the activity of trans-1 against other strains, which would be tested in the future when we have opportunity.

Fig.2 A: The methods section must have qRT-PCR details. What primers were used? 

Response 17: Revised accordingly, line 152-158.

Fig.5 B: How many times the western blot was performed? The internal control of GAPDH is significantly unequal. Hence, a better-quality image or a bar graph with densitometric analysis will be required.  

Response 18: We see the concern of the reviewer. Indeed, the internal control of GAPDH is not so equal and we did the experiment only once. However, Fig. 5B (Western blot) and Fig. 5C (IFA) were relative quantification experiments to support the absolute quantitative qPCR result of Fig. 5A. All the data confirmed that trans-1 mainly inhibits virus infection at the post-entry stage.

Reviewer 3 Report

This study develops two novel heterocylic compounds, among which trans-1 is more potent to be anti-SARS-CoV-2 in both Vero and Calu-3 cells, overcoming the shortness of CQ ineffectiveness in Calu-3 cells. The data are convincing at in vitro levels, and the figures are of good quality. Some concerns should be solved additionally. 

1.      Trans3-9 was shown in Fig.2A, which should be described in the text.

2.      The potency measurement is different if one compares Fig.2A and Fig. 4A. Fig.2A is 100% inhibition, while Fig. 4A is the absolute virus titer. Does that mean trans-1 is more potent in Calu3 than Vero? Or do the authors think trans-1 works independently of cell types and viral entry pathways? The data could make it more clear to be comparable.

3.      Again, with the same concern, Fig. 5A is loading 20uM trans-1. Will there be differences between cell types if given a less effective dose?

4.      The authors better discuss why the modification on trans-1 escapes the CQ’s target in the endocytosis pathway.

Author Response

This study develops two novel heterocylic compounds, among which trans-1 is more potent to be anti-SARS-CoV-2 in both Vero and Calu-3 cells, overcoming the shortness of CQ ineffectiveness in Calu-3 cells. The data are convincing at in vitro levels, and the figures are of good quality. Some concerns should be solved additionally. 

Response 19: We thank the reviewer very much for the positive comments and valuable suggestions.

  1. Trans3-9 was shown in Fig.2A, which should be described in the text.

Response 20: The general structures of compounds trans-3-9 are now included in Supplementary Figure 7, and a brief description was added in line 214-215.

  1. The potency measurement is different if one compares Fig.2A and Fig. 4A. Fig.2A is 100% inhibition, while Fig. 4A is the absolute virus titer. Does that mean trans-1 is more potent in Calu3 than Vero? Or do the authors think trans-1 works independently of cell types and viral entry pathways? The data could make it more clear to be comparable.

Response 21: In fact, trans-1 shows similar efficacy in Vero E6 and Calu-3, as the EC50 values were 3.15 μM (Fig. 3A) and 2.78 μM (Fig. 4C), respectively. Because the screening of trans-1 to trans-9 were not performed in the same one test, the absolute virus progeny copies differ a little bit even for DMSO control groups in different batch of tests, we presented the antiviral activity by normalized progeny production (%).

  1. Again, with the same concern, Fig. 5A is loading 20uM trans-1. Will there be differences between cell types if given a less effective dose?

Response 22: Fig. 5 is a mechanism study. We used drug concentration of > EC99 (17.83 μM and 10.31 μM for Calu-3 and Vero E6, respectively). A high dose (20 μM) was used to achieve complete inhibition of virus infection so that the mode of action of the drug would be clear. We didn’t test lower drug dose, but in our opinion, similar results would be expected.

  1. The authors better discuss why the modification on trans-1 escapes the CQ’s target in the endocytosis pathway.

Response 23: The newly synthesized and tested compounds trans-1 and trans-2 were originally designed to mimic the effect of CQ to elevate the pH of acidic intracellular organelles. However, based on our research, trans-1 may have different antiviral mechanism from CQ, at least in human lung cells.

As can be seen from Figure 1 and S7, these novel compounds are structurally distinctive from CQ and HCQ, although they also possess basic nitrogen atoms, part of tertiary amino groups. The detailed antiviral mechanism of these compounds required further investigations.

Round 2

Reviewer 2 Report

The authors have made considerable changes, and the quality of the manuscript is significantly improved.